# Paper Doped with Polyacrylonitrile Fibres Modified with 10,12–Pentacosadiynoic Acid

**DOI:** 10.3390/ma14144006

**Published:** 2021-07-17

**Authors:** Elżbieta Sąsiadek, Konrad Olejnik, Marek Kozicki

**Affiliations:** 1Department of Mechanical Engineering, Informatics and Chemistry of Polymer Materials, Lodz University of Technology, 90–543 Lodz, Poland; 2Centre of Papermaking and Printing, Lodz University of Technology, 93–005 Lodz, Poland; konrad.olejnik@p.lodz.pl

**Keywords:** functionalized cellulosic material, paper, ultraviolet radiation sensor, security system, security fibres, modified polyacrylonitrile fibres

## Abstract

This work reports a modification of a fibrous cellulose material (paper) by the addition of polyacrylonitrile (PAN) fibres doped with 10,12–pentacosadiynoic acid (PDA). The fibres are sensitive to ultraviolet (UV) light. When the paper containing PAN–PDA is irradiated with UV light it changes colour to blue as a consequence of interaction of the light with PDA. The colour intensity is related to the absorbed dose, content of PAN–PDA fibres in the paper and the wavelength of UV radiation. The features of the paper are summarised after reflectance spectrophotometry and scanning microscopy analyses. All the properties of the modified paper were tested in accordance with adequate ISO standards. Moreover, a unique method for assessing the unevenness of the paper surface and the quality of printing was proposed by using a Python script (RGBreader) for the analysis of RGB colour channels. The modification applied to the paper can serve as a paper security system. The modified paper can act also as a UV radiation indicator.

## 1. Introduction

Dynamic development of the industrial production and packaging of products exposed challenges regarding protection of copyrights and forgery of products. Modern security systems enhance the quality control and protection against illegal and uncontrolled trade of products. Securing products and packaging is the basic measure related to the quality of raw materials, correct application of production processes, determining the conditions of storage, transport, and distribution of products.

In the case of printed documents, the following forms of security can be highlighted: (i) overt—security on a product, visible and available to users; (ii) certified—a security set that is a trade secret and confirmed by a manufacturer for a specific group of users and (iii) hidden—a security set that can only be verified by certification laboratories without providing information about them to the user. Many markers are used for product protection against undesirable impact factors: changes in humidity and temperature, influence of oxygen, or exposure to light, mainly to ultraviolet (UV) radiation. Such markers provide information on climatic, biological, and mechanical factors, including information about product features such as quantitative losses, changes in shape, consistency, taste, and loss of functionality [1]. Absorbing systems, interacting with oxygen, water, odour, carbon dioxide, and ethylene, are also known [2]. On the commercial market the following are also available: TTI (Time and Temperature Indicators), MarkTM (3M), OnVu™, and CoolVu™ (Freshpoint) markers, which provide information on temperature differences in relation to the optimal value specified by the manufacturer. Such indicators are produced on the basis of thermochromic compounds that change their colour depending on temperature [3].

For paper products, the strongest types of security (i.e., the most difficult to counterfeit) are those that are incorporated directly into the paper structure [4,5,6]. Manufacturers of various products use other security methods, such as watermarks, security threads, micro–printed strips, coloured fibres visible in reflected light, white or transparent synthetic fibres glowing under the influence of UV radiation, sequins, and special chemical additives. Some types of paper products are also secured by more common printing or drawing techniques such as a Bar Code, Quick Response Code (QR) or even printed antennas for Radio Frequency Identification (RFID)) or near–field communication (NFC) device [7]. Security methods related to the modification of the paper structure are mainly used for secure documents, banknotes, and excise stamps whereas QR or RFID techniques are increasingly appearing in packaging, e.g., pharmaceutical, cosmetics, textile, dyeing industry, and many others [8]. Printed products and packaging can be protected by the use of special inks, dyes, fillers, and paints, e.g., fluorescent, luminescent, IR-absorbing, photochromic, thermochromic, penetrating, optical variable, iridescent, and chemically reactive paints [6,9,10,11,12]. Another group of markers is photochromic indicators, which are designed for UV radiation monitoring. They provide information in a simple, fast, and visible way without the need for additional measuring devices. Usually, they are used in the form of fibres of various lengths and colours which are mixed with the pulp during papermaking. The distribution of these fibres on the paper surface is random, but they can be concentrated in certain areas. For this purpose one can use (i) coloured fibres visible in reflected daylight, invisible under UV radiation, (ii) coloured fibres visible in reflected daylight and changing their colour upon UV light, and (iii) colourless fibres visible only in UV light. Depending on the customer’s needs and the degree of product protection, various combinations of these fibres blends are used. The most popular security forms of paper are (i) fibrous composition of paper or chemical composition of other substrates; (ii) watermarks; (iii) laser engraving; (iv) marbled paper; (v) security thread; (vi) coloured fibres; (vii) synthetic or artificial fibres glowing under the influence of UV radiation, (viii) sequins, and (ix) chemical indicators.

In recent times, we proposed radiochromic polyacrylonitrile (PAN) fibres doped with 10,12–pentacosadiynoic acid (PDA). The PAN–PDA fibres were characterised in terms of their radiation dose response, related to PDA transformation and morphology [13,14,15,16]. The fibres change colour after exposure to ionising or UV radiation. The colour change results from the properties of PDA, which can exist in two forms: red and blue, where a transformation from one form to another is likely upon some inducements [17,18,19,20]. The developed fibres can also be used as markers, e.g., for indicating the originality of textiles and clothing [13]. In this work, however, we report for the first time the manufacturing of a composite material made from a cellulosic fibres and PAN–PDA fibres. Such modification may be used as an originality marker, e.g., for paper products and packaging, or as an indicator of a product exposure to UV radiation. Basic features of the modified paper are presented.

## 2. Materials and Methods

### 2.1. Preparation of Samples

The polyacrylonitrile microfibre (PAN) was prepared from a co–polymer of poly(acrylonitrile–co–methyl acrylate) (PAN, Zoltek, Nyergesújfalu, Hungary) with 10,12–pentacosadiynoic acid (PDA, Sigma–Aldrich, Saint Louis, MI, USA) according to a procedure described elsewhere [15]. PDA was mixed with the PAN solution in DMF during the preparation of a spinning solution. For the fibre preparation, a 23% (*w*/*w*) solution of PAN in DMF was used, and the concentration of PDA was set to 1% (*w*/*w* of PAN). The microfibres were obtained as a multifilament—a bunch of 250 microfibres. After spinning, the fibres were cut into lengths of 3 mm and incorporated into cellulosic paper pulp. Bleached softwood kraft pine pulp (BSK) was used to prepare laboratory handsheets. The pulp parameters were as follows: initial moisture content: 6.37%, α–cellulose content: 86.6%, DP: 1081, initial Schopper–Riegler value: SR–12. Pulp was refined to SR–30 value. Different mixtures of refined cellulosic pulp with the addition of 0, 5, 10, and 20% PAN–PDA fibres were made, and an additional batch of a mixture of cellulosic pulp and 20% PAN–PDA fibres was refined together. All paper samples were prepared according to ISO 5263–1 standard [21], and one batch of 20% PAN–PDA fibres was refined together with cellulosic pulp to SR–30 value in a PFI mill according to TAPPI T 248 standard [22]. Drainability of refined pulps was tested in accordance with ISO 5267–1:1999 [23]. The laboratory sheets of cellulosic pulps with dopants of basis weight 65 g/m^2^ and average thickness of 130 µm were formed in Rapid–Köthen apparatus according to ISO 5269–2:2005 standard [24].

### 2.2. Basic Features of Modified Paper

The prepared paper samples were conditioned according to ISO 187:1999 standard [25]. All the properties of such papers were tested in accordance with adequate ISO standards: (i) basis weight (Laboratory scale AS 110.R2, Radwag, Radom, Poland, ISO 536:2019) [26], (ii) thickness (L&W Thickness Tester, Kista, Sweden ISO 534:2011) [27], (iii) tensile index (Instron 5564, ISO 1924–2:2008) [28], (iv) Elmendorf tear resistance (L&W Tearing Tester, Kista, Sweden, ISO 1974:2012) [29], (v) Bendtsen roughness (MI Testing Machines Inc., New Castle, DE, United States, ISO 8791–2:2013) [30], (vii) Bursting strength (L&W Bursting Tester, Kista, Sweden, ISO 2758:2014) [31].

### 2.3. Irradiation of Samples

All paper samples were irradiated in UV–curing cabinets (UVP, Upland, CA, USA) to deliver doses up to 10 J/cm^2^ at three wavelengths corresponding to UVA, UVB, and UVC. The instruments were equipped with five UV lamps of the following characteristics: UVA (8/W, type F8T5 black light, 369 nm, Hitachi, Tokyo, Japan), UVB (8/W, type G8T5E, 306 nm, Sankyo Denki, Tokyo, Japan) and UVC (8/W, type G8T5, 253.7 nm, Sankyo Denki, Tokyo, Japan). A given UV dose (J/cm^2^) was delivered automatically using a built–in detector and the control system of the device. All samples were irradiated at room temperature (23 °C).

### 2.4. Spectrophotometric Measurements

Paper–PAN–PDA samples before and after irradiation were measured with a Spectraflash–light reflectance instrument (Spectraflash 300, DataColor, Lawrenceville, NJ, USA). The device was calibrated before the measurements. The light in the range of 190–400 nm was cut off in order to prevent exposure of the samples to UV light during the measurements. The samples were measured immediately after irradiation as well as post-irradiation in order to analyse their stability over time of storage. During storage, the samples were covered with aluminium foil to protect them against daylight. The measurements lead to the reflectance spectra (R [%], 400‒700 nm with 10 nm resolution). Following this, a wavelength for which the change in reflectance was substantial was selected and analysed vs. absorbed dose for further analysis of the paper–PAN–PDA samples’ dose–response characteristics.

### 2.5. Scanning Electron Microscopy Measurements

The morphology of paper–PAN–PDA samples was analysed with the aid of a TESCAN VEGA3–EasyProbe (TESCAN Brno, s.r.o., Brno, Czech Republic) scanning electron microscope equipped with VEGATG software (high vacuum mode (SE); accelerating voltage 7 kV). Before the measurements, all samples were coated by Au–Pd layers using a Cressington Sputter Coater 108 auto system (Watford, UK).

### 2.6. Paper Surface Unevenness

In order to assess the inhomogeneity of the paper surface and the possibility of printing on it, black text and graphical gradient patterns were printed on the samples. The prints were made with a digital inkjet printer (Epson L310, Nagano, Japan; Micro Piezo printhead; 5.760 × 1.440 dpi; CMYK printing system: dye-based inks). The printouts were made with the following parameters: document type, standard paper type, high print quality. In the next step, all samples were measured with an Epson Perfection V750 Pro scanner (Japan, Nagano; cold cathode fluorescent lamp; optical resolution Main 6.400 dpi × Sub 9.600 dpi; 48 bit/colour). The scanning was performed in the reflection mode and the colour depth was 24-bit RGB. All paper samples (30 × 30 mm^2^) were scanned at a resolution of 300 dpi. Other scanning parameters such as brightness, colour saturation correction, colour regulation, and sharpness were switched off and were not taken into account in this study. Selected scans were used to calculate profiles across the paper samples. This was conducted with the aid of a prepared script for reading RGB channels (RGBreader; Python Script with Python Imaging Library; DosLab; [16]). Each sample was depicted using a three-colour RGB scale. For the channel that showed the highest changes in RGB values (from 0 to 255), further calculations were made. For the irradiated paper samples doped with PAN–PDA fibres, the green channel was selected. A detailed description of the RGBreader script is described elsewhere [14]. The obtained results of the analysis of the doped paper were compared with a commercial printer paper (POL Speed, Poland, Baruchowo; 80 g/m^2^). Optical microscopy was performed (Biological microscope, NIKON, Japan, Tokyo) under a magnification of ×10 for the evaluation of the paper surface unevenness.

## 3. Results

### 3.1. Dose Response of Paper–PAN–PDA Samples

All prepared papers doped with PAN–PDA fibres were UVA, UVB, and UVC irradiated at doses ranging from 0 to 10 J/cm^2^. The samples were first inspected with the naked eye, followed by spectrophotometric light reflectance measurements. A view of the samples as seen with the naked eye is presented in Figure 1. After UVB and UVC irradiation all doped paper samples changed their colour from white to blue. By exposure to UV light, blue PAN–PDA fibres emerged from the white paper sheets. The colour intensity was dose dependent; the higher the dose, the higher the intensity of the colour obtained. For all examined samples it was visible that they responded to UVC radiation at lower doses than for UVA and UVB irradiation.

The development of colour was related to the absorbed dose of UV light and concentration of PAN–PDA dopant in the paper structure. Therefore, we assessed the samples quantitatively through the measurements of light reflectance in relation to the absorbed UV dose and concentration of the PAN–PDA dopant. The colour change of the paper samples without the PAN–PDA dopant was also analysed in the range of 0−10 J/cm^2^ (Figure 2a,b). It is known that a long-term exposure to UV radiation can degrade various materials, including paper. The papers exposed to light for a long time change their optical, mechanical, and physicochemical properties. The most visible sign of paper ageing is the change of its optical characteristics (yellowing or bleaching), which is associated with degradation of lignin, cellulose, and hemicelluloses after exposure to light, heat, radiation, pH, and oxidation processes at elevated temperature in the presence of atmospheric oxygen. A slight change in the value of the light reflectance (Figure 2a) can be observed at 400−500 nm, which may result from the UV radiation exposure to 10 J/cm^2^. For instance, the difference in reflectance values at 430 nm for non–irradiated and 2 and 10 J/cm^2^ irradiated paper samples was –1.29, –4.56 and –2.99, –5.72 and –3.63, –5.39%, for UVA, UVB, and UVC, respectively. In Figure 2b, the reflectance relations vs. absorbed UV dose are shown. Linear regressions were adapted to the experimental points. The extracted slopes from these regressions are as follows: –0.0001, –0.0003, –0.0003% cm^2^/J, for UVA, UVB, and UVC, respectively. This denotes that the paper samples without the PAN–PDA dopant are only slightly vulnerable to colour change after irradiation with UVA, UVB, and UVC. However, after irradiation, the samples were not yellow to the naked eye; no significant changes in the range of 550−600 nm were observed. Therefore, it was concluded that the applied UV doses slightly affected the non-modified paper samples in a blue range of Vis light.

The colour intensity increased with radiation dose and depended on the content of the PAN–PDA dopant (Figure 3). In Table 1, CIE Lab parameters of the colours of the UV-irradiated-doped paper samples are presented. CIE Lab parameters specify a colour using a 3-axis system. Accordingly, a* and b* are colour directions: +a* is the red axis, –a* is the green axis, +b* is the yellow axis, –b* is the blue axis, and L* is the lightness axis. The reflectance for the samples decreases between about 500 and 660 nm (Figure 3a,c,e). Regardless of the amount of added PAN–PDA fibres to the paper pulp, all paper samples are almost insensitive to UVA irradiation. As a consequence, exposure of the fibres to UVA did not cause colour saturation, even at high doses of over 10 J/cm^2^. In the case of the UVB irradiation of the paper–PAN–PDA, the reflectance versus dose relations at 500 and 600 nm corresponded to blue colour formation of PDA in the PAN fibres (Figure 3b,d,f).

Upon UVB irradiation, blue colour saturation occurs at doses of 2–2.5 J/cm^2^. The sensitivity of the samples is higher than that for UVA light. Low sensitivity of the samples to UVA radiation is advantageous from the point of view of accidental exposure of the samples to daylight. However, the paper–PAN–PDA samples showed the highest colour change after UVC irradiation. If the paper–PAN–PDA samples were UVC irradiated, the saturation of the blue colour occurred at about 1 J/cm^2^. The dose sensitivity (expressed as a slope of quasi-linear part of the reflectance vs. dose relation) was −0.0074 (5% PAN–PDA dopant), −0.0117 (10% PAN–PDA dopant), and −0.0236 (20% PAN–PDA dopant), i.e., 23, 11, and 10% higher, respectively, than the dose sensitivity for UVB radiation. Elementary characteristics of all paper samples doped with PAN–PDA fibres including threshold dose, measuring range, dose sensitivity, and R^2^ coefficient were determined and are presented in Table 2.

The samples made of refined cellulose pulp had similar sensitivity to UVC radiation as the unrefined samples (Figure 4). After analysis of the CIE Lab values obtained from the measurements of light reflectance at 500 nm for refined and unrefined samples, it can be concluded that the intensity of the blue colour for the refined paper visible after UVC irradiation (especially for the dose range 0.5–10 J/cm^2^) was about 5% lower than that for the unrefined sample.

### 3.2. Long-Term Stability of the Doped Paper

The stability of the paper samples with and without the PAN–PDA fibre dopant irradiated to different doses of UVA, UVB, and UVC was assessed by spectrophotometric light reflectance measurements (Figure 5). The measurements were performed for one month. From the results obtained it was concluded that the fibres were stable, independent of the absorbed dose. All samples kept in a dark place did not change during the storage period. Figure 5 shows example measurements of the paper samples with 20% PAN–PDA dopant irradiated with UVC light in the range of 0−10 J/cm^2^. These observations for UVC light-irradiated samples are analogous to those for UVA- and UVB-irradiated samples. On this basis, it was concluded that the samples both non-irradiated and after UV irradiation are very stable during storage. This feature is important if the doped paper is to serve as a radiation sensor/indicator or if a permanent colour effect is required for marking the originality of a paper product.

### 3.3. Morphology and Mechanical Properties of Paper–PAN–PDA Samples

Manufacturing a paper sheet requires several steps, and many factors affect the properties of the final paper. The choice of raw material composition, degree of refining, types, and amounts of additives influence the final product. The most commonly used additives include sizing agents, fillers, dyes, optical brighteners, additives improving the strength of the paper web, and special additives, e.g., natural or synthetic fibres with special properties. In our experiment, two types of paper were produced: a mixture of refined cellulosic BSK pulp with the addition of 0, 5, 10, and 20% PAN–PDA fibres and a mixture of cellulosic BSK pulp refined together with 20% PAN–PDA fibres. The micro-and macro-structure of the paper plays important roles in the mechanical properties. Thus, the morphology of the paper was evaluated with scanning electron microscopy for non-irradiated, UVA, UVB, and UVC irradiated papers without and with PAN–PDA fibres. The results of the scanning electron microscopy measured papers are presented in Figure 6 for non-irradiated and 5 J/cm^2^ (UVC) irradiated samples. PAN–PDA fibres are marked in blue colour to be more visible in the SEM images. In the case of paper without the addition of PAN–PDA fibres, a typical structure of the natural cellulose paper can be seen in the form of elongated, flattened objects with an uneven surface. PAN–PDA fibres are seen as smooth and round objects incorporated into the structure of cellulose fibres.

Based on the analysis of the elementary parameters of the paper, it was found that tensile index, elongation, tear index, and bursting index decreased as the content of PAN–PDA fibres increased (Table 3). At the same time, the roughness of the paper increased, which is typical for such modifications of paper. However, in the case of the refining of cellulosic pulp with 20% addition of PAN–PDA fibres, it was found that the decrease in the mechanical properties of the paper was lower than that in the case of paper with the addition of the same amount of fibres that had not been exposed to refining process. This can be explained by the fact that the synthetic fibres were deformed during the refining operation, thanks to which they better fit into the fibrous structure of the paper. These conclusions were confirmed by the apparent density results (Table 3). With an increase in the addition of unrefined synthetic fibres, the apparent density decreased. In the case of the pulp refined together with 20% of PAN–PDA fibres, the value of apparent density was higher—practically the same as for paper without the addition of synthetic fibres. The presence of fibre deformation was also confirmed after analysis of Figure 7d.

The mechanical mixing of the PAN–PDA fibres with the cellulose pulp resulted in a homogeneous paper surface and did not significantly change its structure. The straight lengths (3 mm) of PAN–PDA fibres penetrate into the cellulose pulp structure at various depths. After preparation of the doped paper, the samples containing PAN–PDA fibres showed no changes in optical properties and the surface texture appeared uniform. Mechanical mixing of the cellulose pulp ensured homogeneous distribution of PAN–PDA fibres. However, when irradiated with higher doses of UV radiation, bunches of blue fibres were visible on the paper surface (Figure 7). PAN–PDA fibres were distributed randomly, and their visibility on the surface increased with an increase in their amount. Mechanical mixing of the fibre dopant did not damage them. They were straight and smooth without visible changes such as bends, fractures or fibrillation.

To improve the uniformity of PAN–PDA fibre dispersion, refined paper samples were prepared. Refining is required in order to improve papermaking ability and paper properties, among others the structure, flexibility, and strength. Refined pulp was more homogeneous and filled, which is related to the fibrillization of cellulose fibres (Figure 7b). As a result of refining, PAN–PDA fibres were tangled and better bonded to the paper surface (Figure 7d). However, they were not damaged. There were no visible cracks or fibrillation of PAN–PDA fibres. The obtained samples were smoother and the fibre bunches were hardly seen on the paper surface with the naked eye after UV irradiation.

### 3.4. Evaluation of Paper Surface Unevenness

Based on the analysis of the basic parameters of the paper, the surface unevenness of the manufactured papers was examined in relation to the commercial ones. For this purpose, the RGBreader script described in Section 2.6 was used, which enabled the graphical visualization of the sample in the RGB colour space. Each paper sample was imaged with a flat-bed scanner and saved as an bmp file. Then, the image was analysed with the RGBreader script, which decomposed the graphics file into RGB channels. For the paper samples, significant changes in RGB values were recorded in the green channel. Therefore, it was selected for further analysis (Figure 8). The paper with the addition of 10% PAN–PDA fibres showed the greatest surface unevenness. Moreover, the refining of cellulosic pulp with PAN–PDA fibres led to a more filled (less uneven) surface of the paper (Figure 9).

The developed paper is intended to be used as a substrate for inkjet digital printing, e.g., as packaging. Therefore, it was interesting to see if the samples could be printed similarly to the commercial printer paper. It is known that the print quality depends on the basis weight, roughness, structure, and surface properties of the paper. Chemical and physical properties of the printed surface play an important role in paper interaction with liquids. Commercial printer papers are enhanced by the addition of surface sizing to control the wetting and penetration of the liquid. The behaviour of ink drops on the paper surface depends on the surface and interface tension and thus the wetting conditions and the flow of ink during the wetting process. In the digital printing process, the colorant part of the ink fixes in the top layer of paper and the ink vehicle part penetrates deeper into the structure [32].

In assessing the quality of inkjet digital printing, the most important factors are colour reproduction and print sharpness. Assessment of print quality is usually subjective and based on experience with printing. The instrumental measurements must relate to the subjective perceptual impressions to be meaningful. Therefore, a method for assessing the print quality using RGBreader script has been proposed. For this purpose, a text and a graphic pattern (black and colour) were inkjet printed on a commercial printer paper and refined paper doped with 20% PAN–PDA fibres (Figure 10). The naked eye analysis of the samples indicated a visible difference between the saturation of printed colours for commercial printer paper and the paper doped with 20% PAN–PDA fibres. In the case of commercial paper, the colour and black prints were less intense (Figure 10a,d,g), which may be related to the basic parameters of paper. For paper doped with PAN–PDA fibres, disturbances in the green channel caused by the paper surface structure were visible, but they did not significantly affect the shape and sharpness of the printed patterns (Figure 10b,e,h). Moreover, the samples looked similar after UVC irradiation (Figure 10c,f,i). The colour change of the paper was related to the PAN–PDA fibre dopant, but had no effect on the printed pattern. It was found that after UV irradiation, the print quality of the refined paper with 20% PAN–PDA fibres was almost the same as that for commercial paper due to the colour intensity and the mapping of the printed shape (Figure 11a). It was impossible to determine significant differences in the structure of the paper or the quality of prints following the analysis of the photos from an optical microscope (Figure 11b,c).

Additionally, SEM measurements were performed and the obtained results are presented in Figure 12. As expected, no traces of dye or aggregation of ink particles were visible on the surface of the papers. However, the commercial printer paper had fillers in its structure to even out its surface, which was not the case for the paper modified with PAN–PDA fibres.

## 4. Conclusions

In this work we reported the first results from the preparation and characterization of papers modified with PAN–PDA fibres. Preliminary studies have shown that PAN–PDA fibres can be used as a modifier for paper products. Modified papers respond to UV radiation by colour change. The intensity of the blue colour increases with increasing dose of UVB and UVC light. The micro- and macro-structure of the papers depends on the preparation method applied. It was also found that carrying out the joint pulp refining with the addition of the PAN–PDA fibres resulted in a smaller decrease in the mechanical properties of the paper. The refined synthetic fibres were not damaged to such an extent that they lost their optical properties. Moreover, it has been shown that the developed refined paper with 20% PAN–PDA fibre is suitable as a substrate for inkjet printing and the print quality is comparable to that of a commercial printer paper. An application of the modified papers would be marking products against copying and counterfeiting, e.g., packaging with markers for the pharmaceutical industry. The proposed method provides uniform distribution of fibres in the product structure and a smooth surface that may be printed. An advantage of the solution presented is an intensive development of a colour when exposed to UV radiation of a specific wavelength. This can be used as a radiation “on–off” marker and effective identifier of paper products, documents, banknotes, and securities. This can also be helpful in management, control of production, transport, and storage of products. The subject of manufacturing paper modified with PAN–PDA fibres is not exhaustive and requires further research including (i) light properties (fluorescent and absorption spectra) of the papers [33,34,35], (ii) the influence of additives on the surface quality of the paper and UV dose response of PAN–PDA fibres dopant, (iii) the influence of paper refining methods: plasma, high temperature, calendaring.

## Figures and Tables

**Figure 1 materials-14-04006-f001:**
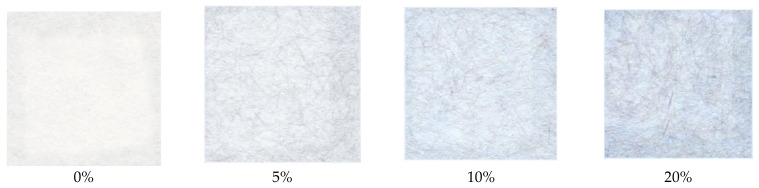
Comparison of paper samples irradiated with the dose of 5 J/cm^2^ of UVC light with different contents of the PAN–PDA dopant (non-refined paper).

**Figure 2 materials-14-04006-f002:**
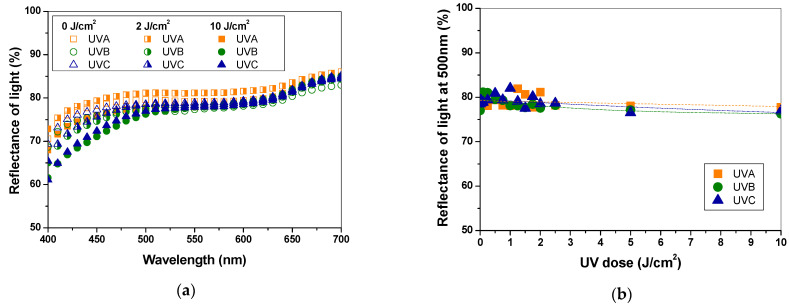
Basic characteristics of paper samples (non-refined paper) without the PAN–PDA dopant irradiated with UV light. (**a**) reflectance spectra measured immediately after UV irradiation (UV doses are given in the graph). (**b**) reflectance (at 500 nm) dependencies versus absorbed dose of UVA, UVB, and UVC light.

**Figure 3 materials-14-04006-f003:**
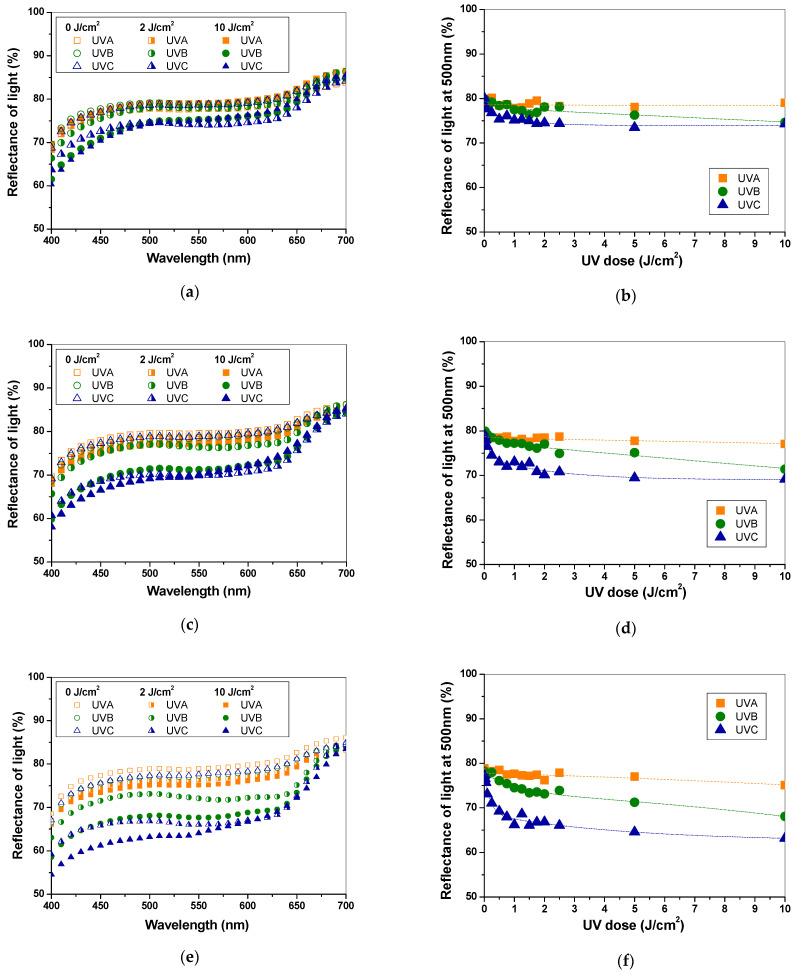
Basic characteristics of paper samples (non-refined paper) with 5% (**a**,**b**), 10% (**c**,**d**), and 20% (**e**,**f**) PAN–PDA dopant irradiated with UV light. Left column (**a**,**c**,**e**): reflectance spectra measured immediately after UV irradiation (UV doses are given in the graph). Right column (**b**,**d**,**f**): reflectance (at 500 nm) dependencies on the absorbed dose of UVA, UVB, and UVC light.

**Figure 4 materials-14-04006-f004:**
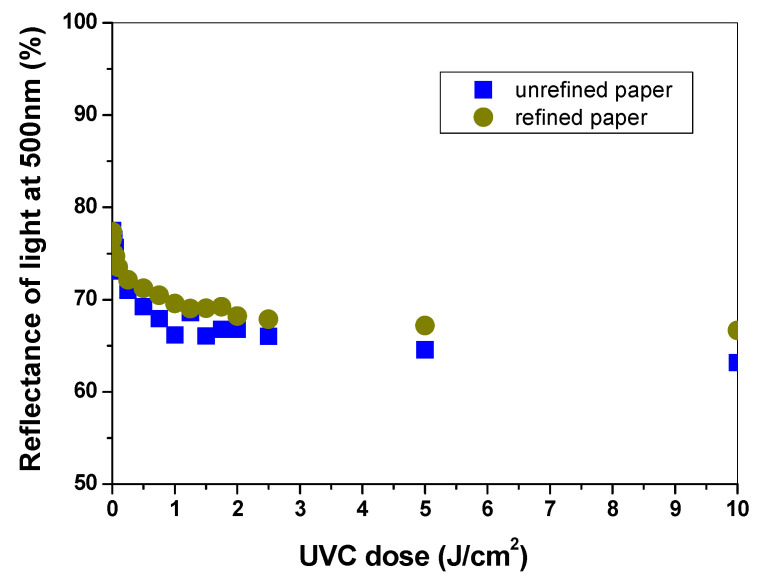
Comparison of the dose response for refined and unrefined paper samples with 20% PAN–PDA fibre dopant made of refined and unrefined cellulose pulp.

**Figure 5 materials-14-04006-f005:**
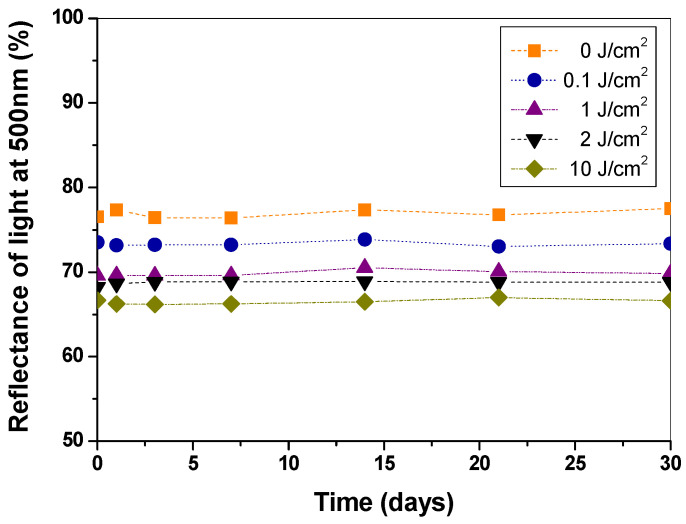
Long-term stability of paper doped with 20% of PAN–PDA fibres after irradiation with UVC light in the range 0−10 J/cm^2^ (non-refined paper).

**Figure 6 materials-14-04006-f006:**
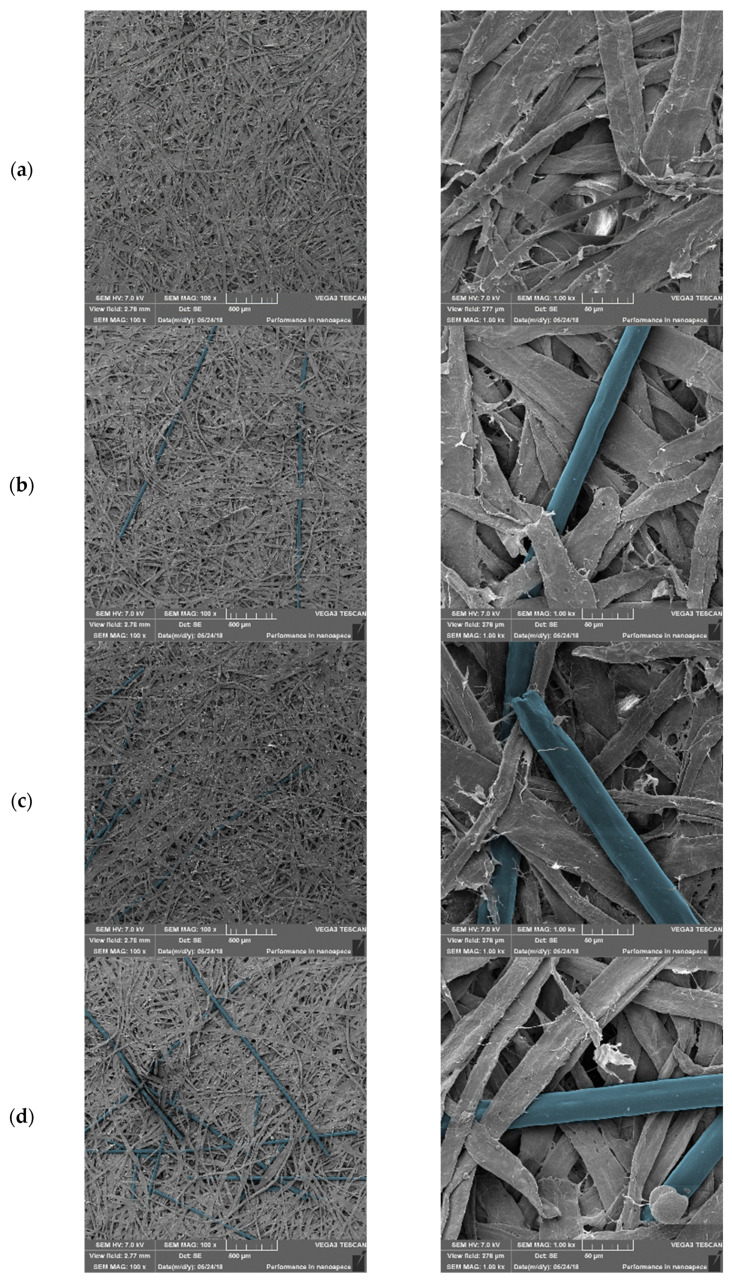
Comparison of the morphology of the paper samples containing 0 (**a**), 5 (**b**), 10 (**c**), and 20% (**d**) of PAN–PDA fibres. Left collumn: magnification ×100, inside scale 100 µm) and right collumn: ×1000, inside scale 50 µm (scanning electron microscopy analysis). The sample without PAN–PDA was not irradiated, whereas other samples absorbed the dose of 5 J/cm^2^ UVC.

**Figure 7 materials-14-04006-f007:**
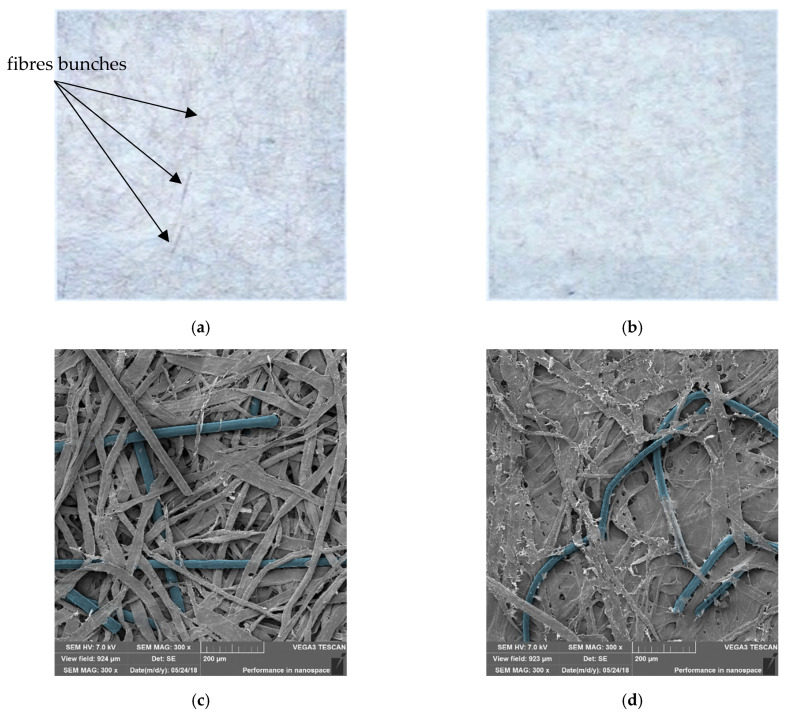
Paper samples doped with 20% of PAN–PDA fibres made of unrefined: (**a**,**c**) and refined: (**b**,**d**) cellulose pulp. Top row: photography of samples with ×10 magnification. Bottom row: SEM images of the same samples with ×300 magnification, inside scale 200 µm. The images show the doped paper after UVC irradiation at 5 J/cm^2^.

**Figure 8 materials-14-04006-f008:**
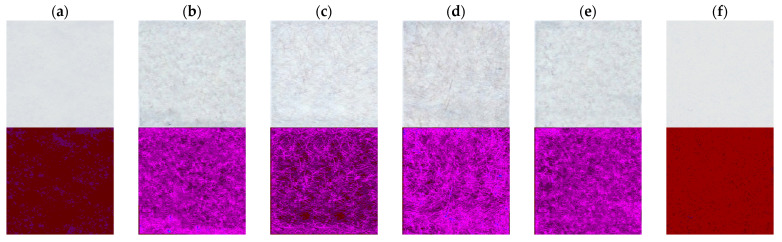
Analysis of the green channel after applying the RGBreader script for paper samples: (**a**) 0% PAN–PDA fibres; (**b**) 5% PAN–PDA fibres; (**c**) 10% PAN–PDA fibres; (**d**) 20% PAN–PDA fibres; (**e**) paper refined with 20% PAN–PDA fibres; (**f**) commercial printer paper. Top row: image of the samples after scanning; bottom row: green RGB channel from RGBreader.

**Figure 9 materials-14-04006-f009:**
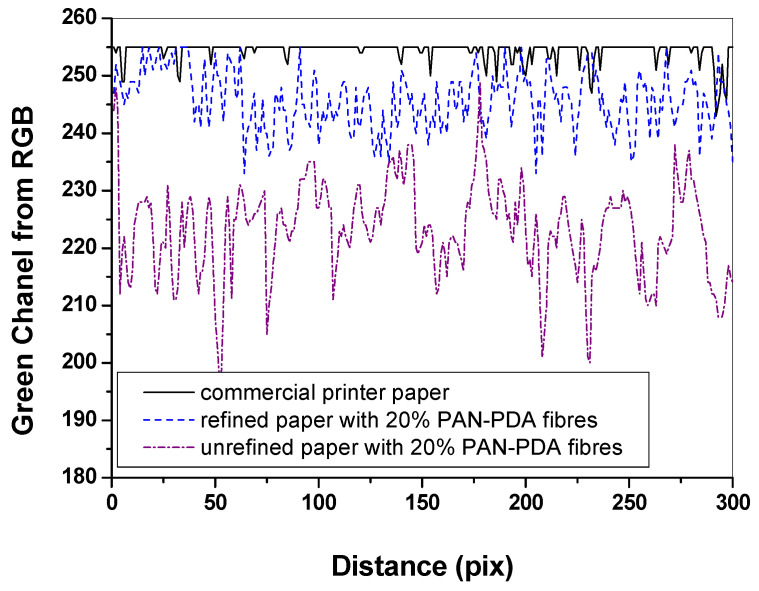
Comparison of surface unevenness of paper samples (commercial paper, unrefined paper with 20% PAN–PDA fibres, and refined paper with 20% PAN–PDA fibres).

**Figure 10 materials-14-04006-f010:**
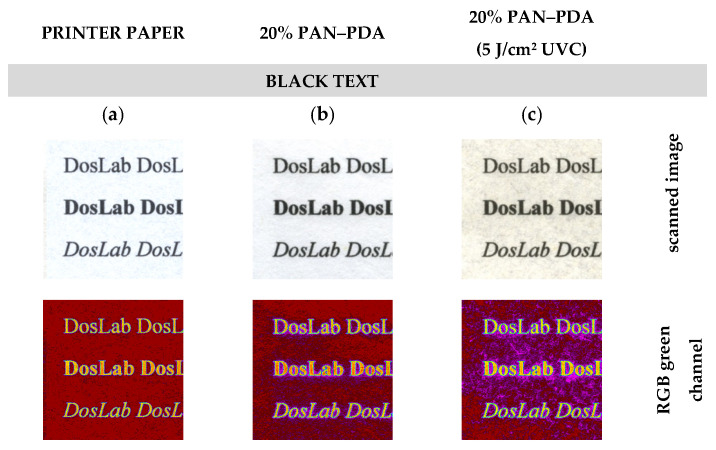
Comparison of printed patterns and print quality on the surface of paper samples: (**a**,**d**,**g**) commercial printer paper; (**b**,**e**,**h**) refined paper with 20% PAN–PDA fibres; (**c**,**f**,**i**) refined paper with 20% PAN–PDA fibres after irradiation (5 J/cm^2^ UVC).

**Figure 11 materials-14-04006-f011:**
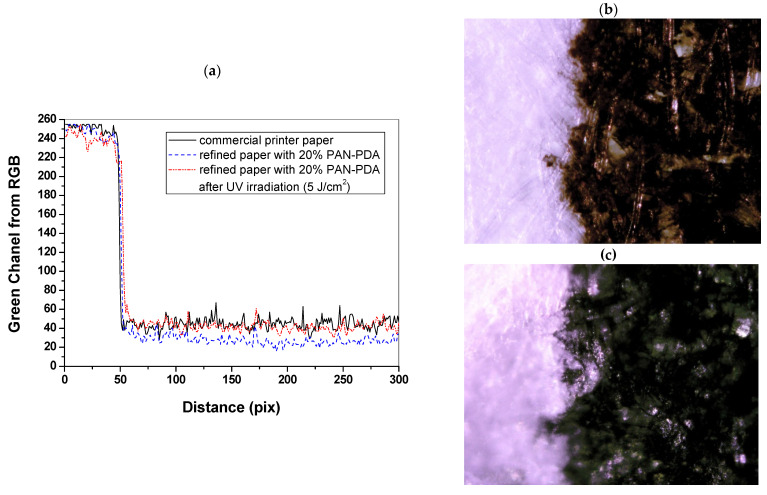
Print quality assessment: (**a**) comparison of print quality based on changes in the RGB green channel value of the samples and images from an optical microscope of the print on (**b**) commercial printer paper and (**c**) refined paper with 20% PAN–PDA fibres; ×10 magnification.

**Figure 12 materials-14-04006-f012:**
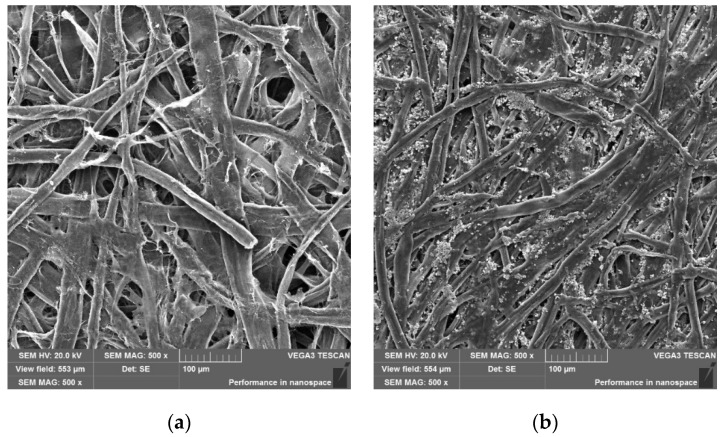
Comparison of the morphology of (**a**) the paper sample containing 20% of PAN–PDA fibres and (**b**) commercial printer paper (magnification ×500; scale 100 µm; scanning electron microscopy analysis).

**Table 1 materials-14-04006-t001:** CIE Lab parameters of paper doped with PAN–PDA fibres derived from reflectance measurements of samples irradiated with UV light (non-refined paper).

Dopant Content (%)	CIE Lab Values	UVA Dose (J/cm^2^)			UVB Dose (J/cm^2^)			UVC Dose (J/cm^2^)		
		0	1	10	0	1	10	0	1	10
0	L	92.41	90.80	90.82	90.53	90.86	90.62	91.03	92.71	90.62
	a	−0.14	−0.11	0.11	0.32	−0.35	−0.51	0.15	−0.42	−0.80
	b	1.04	1.27	1.86	1.40	2.05	5.26	1.17	2.53	4.39
5	L	91.73	90.80	91.32	91.23	90.47	89.50	91.13	89.47	89.49
	a	−0.06	0.17	−0.03	−0.03	−0.31	−0.40	−0.04	−0.32	−0.17
	b	1.36	1.55	1.72	1.17	1.74	3.64	1.55	2.09	4.11
10	L	91.54	90.76	90.54	91.13	90.46	87.76	91.26	88.25	87.23
	a	−0.04	−0.04	0.13	0.07	−0.22	0.03	0.06	0.01	0.98
	b	1.49	1.78	1.90	1.65	1.90	2.46	1.51	0.91	3.60
20	L	91.29	90.80	89.60	90.54	88.93	86.08	90.66	84.74	84.35
	a	0.20	0.15	0.30	0.21	−0.34	0.32	0.14	0.46	1.72
	b	1.56	1.83	1.74	1.72	1.18	1.84	1.84	−0.01	3.34

**Table 2 materials-14-04006-t002:** Elementary characteristics of UV-irradiated paper samples with PAN–PDA fibres dopant (non–refined paper) (A denotes the slope of the linear regression that is the dose sensitivity).

UV Energy	PDA Dopant (%)	Threshold Dose (J cm^–2^)	Dynamic Range (J cm^–2^)	Quasi Linear Range (J cm^–2^)	A (% cm^2^ J^–1^)	A_0_	R^2^
**UVA**	5	–	–	–	–	–	–
	10	2	2–10	–	–	–	–
	20	2	2–10	–	–	–	–
**UVB**	5	0.05	0.05–10	0.5–1.5	–0.0017 ± 0.0004	79.37	0.8941
	10	0.025	0.025–10	0.25–2.5	–0.0013 ± 0.0003	78.61	0.9041
	20	0.025	0.025–10	0.1–2	–0.0025 ± 0.0003	77.65	0.9570
**UVC**	5	0.025	0.025–5	0.025–0.75	–0.0074 ± 0.0015	78.82	0.9108
	10	0.025	0.025–10	0.025–0.5	–0.0119 ± 0.0017	78.45	0.9579
	20	0.025	0.025–10	0.025–0.5	–0.0236 ± 0.0054	76.59	0.9519

**Table 3 materials-14-04006-t003:** Basic parameters of the paper samples doped with different amounts of PAN–PDA fibres (SD denotes the standard deviation and COV denotes the coefficient of variation).

PAN–PDA Fibre Content (%)	Apparent Density (g/cm^3^)	Roughness (mL/min)	Tensile Index (N × m/g)	Elongation (%)	Tear Index (m^2^ × mN/g)	Bursting Index (kPa × m^2^/g)	
0	0.508	315.33	20.32	3.27	3.82	1.22	
	0.004	37.34	0.43	1.16	1.56	2.18	SD
	0.007	0.118	0.02	0.35	0.41	0.03	COV, %
5	0.496	393.22	20.13	2.91	3.42	1.23	
	0.009	39.544	0.82	0.41	0.02	5.27	SD
	0.018	0.101	0.04	0.14	0.01	0.07	COV, %
10	0.490	505.11	16.55	2.85	3.30	1.12	
	0.008	70.113	3.98	0.12	0.03	4.73	SD
	0.016	0.139	0.24	0.04	0.01	0.06	COV, %
20	0.452	613.11	16.17	2.52	2.49	0.98	
	0.009	91.820	0.83	0.50	0.09	3.82	SD
	0.019	0.150	0.05	0.20	0.04	0.06	COV, %
20 *	0.506	427.12	18.78	2.90	3.39	1.21	
	0.009	32.93	0.96	0.27	0.03	0.08	SD
	1.760	7.71	5.13	9.31	0.89	6.62	COV, %

* refined together with cellulosic pulp.

## Data Availability

Data available on request by contacting with the corresponding authors.

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
