# Peer review of "Paper Doped with Polyacrylonitrile Fibres Modified with 10,12–Pentacosadiynoic Acid"

_materials, 2021, doi:10.3390/ma14144006_

Round 1
Reviewer 1 Report
Dear Authors,
This manuscript brought up a very interesting issue regarding the use of PAN-PDA for color changing in the field of paper security system application. However, following issues should be modified before the publishing and the merits of this structure and process need to be further emphasized.
- Introduction: It is well organized and explained, but more explanation of the color-changing fiber and the material used in this study, PAN-PDA, are needed. It would be good to include more references as well.
- (line 176) As the authors mentioned, exposed time is an important parameter for the properties. What is the condition of time and temperature? Is it controlled? And did you investigate their effect? If you do, it is needed a definition and explanation for it.
- (line 193) Following No. 2, only UVB seems to be sensitive to intensity changes. Is this effect related to time? Whether UVA has the same effect if the intensity is increased or the time is increased?
- (line 194) It seems better to briefly describe the CIE Lab parameters.
- (line 224) Did you test pristine PAN-PDA fiber? As the results of 5, 10, and 20 %, the tendency does not seem to be linear, what are the properties of the PAN-PDA material?
- (line 300) Is there any way to further refine the surface? The surface condition is not good compared to commercial paper. Could you use other methods such as heat treatment or a form other than fibrous to improve the surface? There seems to be a quality problem with using paper like this.
- (line 316) Is there any effect of UVC to commercial paper? Also, is there any effect of type of ink?
- Conclusions: Is there any advantage for using fiber structure of PAN-PDA? Compared to particles or surface coating, there seems to be a limit to the surface condition of the paper with the fibrous form. Why is the fibrous form used including additional processes such as fiber spinning and refining? The merits of this structure and process need to be further emphasized.
- The following modifications are required:
- Minor spell check and grammar - such as usage of 'and' and 'equal to'
- High resolution image for Figure 6 and bigger scale bar
- Rearrange Figure 10 for visibility and efficiency
- Uniformity of tables and figures and definition of each variable
- Statistical study for values such as Table 3
Reviewer 2 Report
This paper describes a production of Radiochromic paper products with cellulosic fibers and PAN fibres doped with 10,12-PDA, which is insteresting in the paper-based product industry. It can be accepted after taking consideration of the following comments.
(1) Introduction: The author should extensively review different methods to produce Radiochromic paper products and related previous studies by other people, such as Cellulose 2019, 26, 5117–5131, Nanomaterials 2019, 9, 1322; doi:10.3390/nano9091322 etc.
(2) Fluorescence emission spectra and UV-Vis absorbance spectra of the radiochromic paper should be obtained to evaluate the performance of the paper under different light wavelength especially ultraviolet.
(3) Table 3: these data should have standard error.
(4) The changes of radiochromic properties of the paper with wearness and foldness (durability) of the paper should be evaluated.
(5) This paper has too much content on paper unevenness, printability etc., which is distracting. It should focus more about its light properties.
Round 2
Reviewer 1 Report
The authors confirmed the contents of the revised paper. I think it's okay to be published in this journal.
Reviewer 2 Report
This paper can be accepted.